# Genomic hotspots in the DENV-2 serotype (E, NS4B, and NS5 genes) are associated with dengue disease severity in the endemic region of India

Varsha Ravi[1,2‡], Kriti Khare[1,3‡], Ramakant Mohite[1], Pallavi Mishra[1], Sayanti Halder[1], Richa Shukla[1], Chinky Shiu Chen Liu[1], Aanchal Yadav[1,3], Jyoti Soni[1,3], Kanika[1], Komal Chaudhary[3,4], Neha[3,4], Bansidhar Tarai[4], Sandeep Budhiraja[4], Pooja Khosla[5], Tavpritesh Sethi[2], Md Imran[1]*, Rajesh Pandey◉[1,3]*

**1** Division of Infectious Disease Biology, INtegrative GENomics of HOst-PathogEn (INGEN-HOPE) laboratory, CSIR-Institute of Genomics and Integrative Biology (CSIR-IGIB), Mall Road, Delhi, India, **2** Indraprastha Institute of Information Technology, Delhi, India, **3** Academy of Scientific and Innovative Research (AcSIR), Ghaziabad, India, **4** Max Super Speciality Hospital (A Unit of Devki Devi Foundation), Max Healthcare, Delhi, India, **5** Sir Gangaram Hospital, New Delhi, India

‡ These authors share first authorship on this work.
\* mdimran916@gmail.com (MI); rajesh.p@igib.res.in, rajeshp@igib.in (RP)

## Abstract

Dengue virus (DENV) infection manifests a wide spectrum of clinical outcomes, ranging from mild fever to severe and potentially fatal disease, yet the factors driving this variability remain poorly understood. This study aims to unravel the relationship between clinical manifestations of dengue and the genetic diversity of the virus, providing insights into the genomic variability driving disease severity. To achieve this, serum samples were collected during a dengue outbreak in National Capital Region-Delhi, India, from June to November 2023. Serotyping of RNA isolated from 4,045 clinical serum samples revealed DENV-2 as the predominant serotype in circulation (n = 3702). Whole-genome sequencing for 3702 clinical samples was performed using Oxford Nanopore Technology (ONT) further yielding 3254 DENV-2 genomes with >50% coverage. However, all of them identified the cosmopolitan genotype of DENV-2, forming a distinct monophyletic cluster in the global phylogenetic tree. Comprehensive variant analysis uncovered 1,618,158 single nucleotide variations (SNVs) across the sequenced DENV-2 population. The clinico-genomic approach carried out in 1294 samples, mild (n = 473), moderate (n = 405) and clinically severe (n = 416), reveals a significant burden of SNVs in various genomic regions linked to differential disease outcomes. Statistical analyses, including Fisher's exact test and phi-correlation, identified hotspot regions in the Envelope (E), NS4B, and NS5 genes, where SNVs were strongly associated with mild and clinically severe phenotypes, providing insights into the genomic determinants of disease severity. Interestingly, the clustering of severity-associated SNVs in these genomic hotspot regions highlights their potential

**Data availability statement:** The DENV-2 genomic data has been uploaded to GISAID (EPI IDs are provided in S2 Table) and is available to the global community.

**Funding:** This study received financial support from Bill and Melinda Gates Foundation (BMGF), (Grant number - INV-033578) and Rockefeller Foundation, (grant number - 2021 HTH 018) awarded to RP. VR, MI, RS, CL, and PM received their salary from BMGF. RM and SH received financial support from AIDS Healthcare Foundation (AHF), (grant number - GAP0276). K., received financial support from GIISER, South Asia, (grant number – GAP0248). The funders had no role in study design, data collection and analysis, decision to publish, or preparation of the manuscript.

**Competing interests:** The authors have declared that no competing interests exist.

as therapeutic targets within the DENV genome. These findings offer a promising direction for developing early mitigation strategies and targeted interventions to manage the progression of severe DENV infections.

## Author summary

Dengue has emerged as a significant global public health concern, particularly in tropical and subtropical regions. The escalating rate of DENV infections in endemic areas underscores the urgent need for studies involving larger datasets that integrate viral genomic information with clinical data to better understand patient outcomes and disease severity. Therefore, we conducted a clinico-genomics study using serum samples from dengue patients in the National Capital Region of Delhi, India, with the goal of understanding the evolution and genomic variability of the dominant serotype, DENV-2. By sequencing the whole genome of DENV-2, we identified a wide range of SNVs within the virus. We also captured key clinical features of the patients and integrated this clinical data with genomic information to explore how specific genomic variants may be linked to different disease severities, including mild, moderate, and severe forms of dengue. Our analysis revealed a significant burden of SNVs in DENV-2 genes, with specific hotspots in the E, NS4B, and NS5 regions being strongly associated with clinically severe disease outcomes. This integrative approach provides valuable insights into how viral genetic variations contribute to disease severity and can help improve future dengue surveillance, prevention, and treatment strategies.

## Introduction

Dengue, a disease of global concern, especially in the subtropical regions, is transmitted by *Aedes aegypti* mosquitoes. The occurrence of dengue has steadily intensified, with an estimated 390 million infections annually across more than 100 countries. This is due to many factors such as rapid urbanization, global travel, and climate change that in combination broaden the virus's reach and seasonality [1]. The highest incidence of dengue occurred in 2023, with over 6.5 million cases across 80 countries [2]. The current data of WHO for the year 2024 suggests further escalation, with over 7.6 million cases including 16,000 severe dengue, and 3,000 deaths [2].

Dengue is caused by the dengue virus (DENV) that belongs to the Flaviviridae family, a group containing over 70 pathogenic viruses affecting intertropical regions [3]. The DENV exists in four serologically distinct serotypes (DENV-1, DENV-2, DENV-3, and DENV-4), each with unique immune response dynamics. The genome of DENVs—a positive sense, single-stranded RNA virus—is approximately 10.7 kb, comprising a coding region flanked with a 5' untranslated region (5'UTR) and a 3'UTR[4]. The coding region of DENVs' genome encodes a single polyprotein.

Post-translationally, the polyprotein gets cleaved into three structural proteins and seven non-structural proteins. The structural protein includes capsid (C), pre-membrane/membrane (prM/M), and envelope (E). The non-structural proteins are NS1, NS2A, NS2B, NS3, NS4A, NS4B, and NS5 [4]. The structural proteins of the DENVs primarily mediate virus attachment, entry, assembly, and secretion. The non-structural proteins carry essential enzymatic activities for virus fitness in the host. Specifically, the C-terminal of NS5 carries RNA-dependent RNA polymerase (RdRp) that facilitates replication of the virus genome [5]. More importantly, the RdRp is a low-fidelity replicative enzyme and thus prone to introduce genetic variation in the virus genome during each replication cycle.

The DENV virus infection manifests in a wide range of alterations in haematological parameters of dengue patients. The TLC and platelet count reduction across dengue patients are well documented and widely employed to assess the systemic impact of the infection. Additionally, the TLC and platelet count associated with dengue patients are considered the major predictive parameters of dengue fever (DF) [6–8]. Moreover, prolonged leukopenia and thrombocytopenia have been implicated in the transition from mild to severe dengue, likely due to their role in impairing the host's immune response and exacerbating vascular leakage, bleeding tendencies, and organ failure. Liver is a very common susceptible organ that is known to develop chronic liver dysfunction to liver failure [9] in severe dengue. The liver damage in DENV infection is often attributed to direct viral cytopathic effects and immune-mediated injury [10–13]. The liver damage/dysfunction widely inferred by observing elevated levels of SGOT-Aspartate Transaminase (AST) and SGPT-Alanine Transaminase (ALT) in the blood of the dengue patients and known as the key indicators of liver damage [9,14,15]. Together, leukopenia and thrombocytopenia, along with elevated AST and ALT levels, may indicate severe dengue, reflecting immune dysfunction, vascular complications, and liver damage.

As DENV virus genomic variability plays a key role in the disease outcome and clinical manifestation across dengue patients, to effectively curb the growing threat of dengue, there is an urgent need for advanced, real-time, integrative, clinico-genomic and epidemiological surveillance systems. Using this combined approach is imperative to explore the plausible role of genomic variability in understanding the disease dynamics and progression, especially with the increasing trend of DENV infections globally, and sub-tropical regions specifically. Studies have been conducted to investigate the genomic variability across the DENV virus genomes, focusing on determining the genetic heterogeneity and evolution [16–18]. Moreover, there are few reports that focus on investigating the potential link between the genetic variations in the DENV virus genome and the disease severity [19–22]. For, e.g., the mutation hotpots spanning the 18-basepair region 523–541 in prM/M and 10387–10405 of 3'UTR of DENV genome are associated with severe dengue disease [4]. Furthermore, correlation of DENV mutations with TCR clonotypes reveals that the T10407C mutation in the 3′ UTR is positively associated with TRGV and TRDV exclusively in severe patients [23]. However, the study capturing the genomic variability across a large set of DENV genomes using the clinical samples and evaluating its role in the disease outcome is very scarce in number and scale. Therefore, we employed a clinico-genomics approach and conducted an integrative analysis of genetic variability across the 3254 in-house sequenced DENV-2 genomes and the clinical parameters of dengue patients in this study cohort. Given the high incidence of DENV-2 infections in the Indian subcontinent, we investigated infections by DENV-2 for a deeper understanding of the plausible functional role of the genomic variants in one particular serotype vis-à-vis clinical severity. Our study identifies a core set of single nucleotide variants (SNVs) associated with differential clinical outcomes of the patients and highlights the importance of sustained clinico-genomic surveillance as a way forward to understand the dengue disease severity and strengthening potential pandemic preparedness.

## Methodology

### Ethics statement

The study was designed in accordance with the Declaration of Helsinki and was approved by the institutional ethics committee of the Council of Scientific & Industrial Research-Institute of Genomics and Integrative Biology, Delhi, India (Ref No: CSIR-IGIB/IHEC/2023–24/04). The patients provided their written informed consent before participation in this study.

## Study design and sample processing

The study was conducted by utilising the serum samples collected from the patients reporting to the Max Super Speciality hospital, New Delhi, India, during June to November 2023. The serum samples from symptomatic dengue-suspect patients were collected at the hospital and subjected to an NS1-Ag test. A total of 4045 NS1-Ag-positive serum samples were brought to our laboratory at CSIR-Institute of Genomic and Integrative Biology (CSIR-IGIB), New Delhi, India, in a cold storage box on a daily basis. The transportation time was ranging from 45 to 60 minutes. On the same day, the serum samples were aseptically transferred to the 2 ml tubes and processed for the RNA isolation. and remaining serum samples were stored at -80°C for long-term preservation (in case required). To each serum sample, a unique barcode ID was assigned. The clinical data for a subset of those patients who underwent a complete blood count (CBC) and/or liver function test (LFT) were also obtained from the hospital EHR (Electronic Health Record). Out of 4045 patients from which we have NS1-Ag positive serum samples, 1756 CBC reports and 790 LFT reports were obtained and utilized in this study for the analysis.

Further, for the purpose of analytical clarity, we categorized our patient cohort, based on the local clinical parameter data, into mild, moderate, and clinically severe as reported earlier [23–25]. Following these earlier studies, we have investigated our clinical data for the classification of dengue patients. Across the 1756 CBC reports, 473 patients demonstrated both Total Leukocyte Count (TLC) and platelet count within normal range. Similarly, 405 patients were observed with low TLC and normal platelet count; whereas, 416 patients had both low TLC and low platelet count compared to the normal range. Thus, based on CBC reports of the patients, we were able to categorize the 1294 dengue patients into three groups: dengue without warning signs (n = 473, patients with normal platelet and leukocyte counts), dengue with warning signs comprising two subgroups—n = 405, patients with normal platelet but decreased leukocyte counts (leukopenia), and n = 416, patients with both low platelet counts (thrombocytopenia) and decreased leukocyte counts. The symptoms associated with severe dengue, such as severe bleeding, organ failure, or abnormal liver parameters, were not exhibited by any patients in our study cohort. In this context, the term clinically severe is used solely to describe the concurrent presence of both thrombocytopenia and leukopenia. For analytical clarity, the groups are designated as mild (patients without thrombocytopenia and leukopenia), moderate (patients without thrombocytopenia but with leukopenia), and clinically severe (patients with both thrombocytopenia and leukopenia) dengue cases. These classification criteria have been summarized in Table 1.

## RNA Isolation and determination of DENV serotype

The serum sample was thawed in ice. From the 150 μl serum samples, total RNA was isolated using QIAamp Viral RNA Mini Kit (Qiagen, Cat No. 52906) following the manufacturer's protocol (and stored at -80°C until further use). Nano-Drop (Thermo Fisher Scientific, USA) was used to assess the quality and quantity of the RNA. A fraction of RNA was utilized to determine the serotype of the DENV through RT-PCR. The DENV serotyping was performed using the DENV serotyping kit following the manufacturer's instructions, with positive and negative controls included (TRUPCR; Cat. No. 3B237).

**Table 1. Criteria based on clinical parameters for defining mild, moderate, and clinically severe dengue patients.**

| Group | Total leukocyte count | Platelet count |
|---|---|---|
| Mild (n = 473) | Normal | Normal |
| Moderate (n = 405) | Decreased (leukopenia) | Normal |
| Clinically Severe (n = 416) | Decreased (leukopenia) | Decreased (thrombocytopenia) |

## Library preparation and whole genome sequencing of DENV-2

A total of 3702 DENV-2 identified using serotype-specific RT-PCR was further processed for whole genome sequencing using Oxford Nanopore Technology (ONT). Briefly, the RNA was reverse transcribed into cDNA using the LunaScript RT Supermix (New England Biolabs, Cat. No. E3010L). The DENV-2 genome was amplified using cDNA as a template. For the amplification of the DENV-2 genome, an amplicon-based tiling PCR approach was employed. Briefly, for the amplification of DENV-2 genomes, 39 sets of overlapping PCR primers adopted from Samuel et al., 2020 [26] were used. Primers were synthesized by Integrated DNA Technologies and optimized for accuracy before use. The sequences of PCR primers are listed in the S1 Table. The 39 sets of PCR primers were divided into two pools: the odd set (pool 1) and the even set (pool 2). The PCR was performed in two sets separately using the primers from pool 1 and pool 2. The amplification is achieved by Q5 High Fidelity 2X Master Mix (New England Biolabs, Cat. No. M0494L). Followed by the PCR, pool 1 and 2 PCR products were pooled together for each reaction. The PCR product was purified using Ampure XP (AXP) beads (Beckman Coulter, Cat. No. A63881). The concentration of purified PCR product was determined using a Qubit 4.0 fluorimeter (Invitrogen, USA). The purified PCR products were then subjected to library preparation. End-repair and A-tailing were performed using the NEBNext Ultra II End Repair/dA-Tailing Module (New England Biolabs, Cat. No. E7546L), followed by barcode ligation (Native Barcoding Kit 24v14; SQK-NBD114.24). The barcoded library was pooled together and processed for purification using Ampure XP (AXP) beads. The concentration of purified barcoded libraries was determined using a Qubit 4.0 fluorimeter (Invitrogen, USA). To the purified barcoded library, the native adapter was ligated (Native Barcoding Kit 24v14; SQK-NBD114.24), and the final purification was done using Ampure XP (AXP) beads. The concentration of the adapter-ligated library was determined using a Qubit 4.0 fluorimeter (Invitrogen, USA). The final purified library was loaded onto a flow cell (R10.4.1) and sequenced using a GridION device (ONT), followed by high-quality DENV-2 genome analysis.

## Analysis of sequencing reads, genome assembly and variant calling

Following the completion of the sequencing runs on the GridION device, the raw ONT reads were transferred to high-performance computing (HPC). Briefly, raw reads were basecalled and demultiplexed through dorado basecaller with a phred Q-score quality of > 9. The demultiplexed high-quality passed reads were mapped to the DENV-2 reference (NC_001474) genome using minimap2 [27]. Additionally, primer trimming was carried out using align trim [28]. Further, artic mask [28] was used to mask the low-quality reads and taken as 'coverage mask' files. Primer depleted high-quality DENV-2 reads were taken for variant calling through Clair3 [29]. Post-VCF filtering was carried out using bcftools [30] by removing the SNVs having genotype quality less than 3 and depth less than 20 as 'fail vcf' and taking genotype quality greater than 3 and depth greater than 20 as 'pass vcf'. Lastly, consensus FASTA was created by masking the positions of 'fail vcf' and 'coverage mask' and implementing the SNVs from 'pass vcf' to the DENV-2 reference genome using bcftools [30]. After merging the multiple VCFs, the preprocessing steps involve 'IndelGap' correction and 'multiallelic' variant resolution using bcftools, allowing us to extract a single variant from a single position. To facilitate downstream analysis, we applied variant normalization using 'norm' and 'sort', ensuring proper merging of SNVs and maintaining a consistent positional order from start to end. The depth and breadth of genome coverage were determined for the reads mapping to the DENV-2 wild-type genome (S2 Table). Samples which are greater than 50% genome coverage are merges and annotations for the SNVs obtained using Snpeff are mentioned in S3 Table.

## Assessment of coverage and depth uniformity

We have generated a significantly high number of passed sequencing reads (with a Q score of 9), achieving a highest read of 2,23,095, with average depth of 1081X and an average genome coverage of 87.5 per Dengue virus genome. The highest genome coverage obtained was 98.8, which is commendable for direct clinical sample sequencing at a large

scale. For basecalling and barcode demultiplexing, we utilized the dorado (v7.2) with the "hac" (high-accuracy basecalling) algorithm. This algorithm enhances sequencing accuracy by using a deep-learning model trained to minimize basecalling errors, ensuring high-fidelity read conversion from raw nanopore signals. Furthermore, Variant calling was carried out using Clair3, a deep-learning-based variant caller specifically designed for long-read sequencing. To ensure high-quality variant identification with uniform depth of coverage, we applied a stringent read-depth threshold, masking variants with depth below 20X.

### Integrative analysis of SNVs across DENV-2 genome with clinical parameters across dengue patients

According to the disease sub-phenotype mentioned, the SNVs identified across DENV-2 genomes were categorized as mild, moderate, and clinically severe subgroups (S4 Table). In order to find the SNV burden in the genes, the presence and absence of SNVs in mild vs. moderate, moderate vs. clinically severe and mild vs. clinically severe were calculated (S5 Table). Further, chi-square test was applied on the obtained 'presence-absence' contingency table. Additionally, on the secured burden genes, the nonparametric fisher's exact test was used to evaluate the independence between two categorical variables and to compare SNV profiles across the groups. Two-sided p-values were computed, with a significance threshold of 0.05. The direction and strength of the association between the SNVs and group classifications were assessed using the phi coefficient ($\varphi$). The chi-square and fisher exact tests were performed using the scipy.stats package in python. Phi correlation was performed using a psych library in R software (S6 Table).

### Phylogenetic Analysis and Genotype detection

The DENV-2 reference sequence (NC_001474) was used to perform multiple sequence alignment with our cohort dataset above 90% coverage through Augur [31]. As these are direct sequencing of the clinical samples, we have kept the threshold at 90% coverage, so as not to miss important clinical implications. Prior to the phylogenetic analysis, we downloaded the DENV-2 genomes with a coverage of above 90% from GISAID on September 12, 2024. We obtained a total of 8085 such DENV-2 genomes. These genomes were submitted to GISAID from different countries/continents of the world (S7 Table). Specifically, in our global data set, the DENV-2 genome was represented from India (n = 1040), Africa (n = 556), Asia (n = 2921), the Caribbean (n = 29), Europe (n = 5), North America (n = 1025), Oceania (n = 75), and South America (n = 2934). Moving forward with the filter of continents having more than 100 genomes, which excludes Europe, the Caribbean, and Oceania. A total of 10164 genomes (Cohort Data = 2188, Global Data = 7976) were taken for phylogenetic analysis, using *'iq-tree'* with the *'maximum-likelihood'* algorithm. The tree was visualized using FigTree1.4.4 [32] as a cladogram with reference sequence as midpoints in increasing order nodes.

### Evaluation of global frequency with significant SNVs

Further, to unravel the occurrence and frequency of the significant SNVs associated with disease severity across the globe, we conducted SNV analysis and frequency comparison across the genomes of DENV-2 reported worldwide. From the 7976 DENV-2 genomes, SNVs were called using the in-house Python code, and frequency was determined with respect to the continents. Additionally, comparisons of our statistically significant SNVs associated with mild and clinically severe phenotype were done using matrix plots.

## Results

### DENV-2: A predominant serotype in circulation

RNA was isolated from 4045 serum samples collected from the NS1-Ag-positive patients in the Delhi-NCR (National Capital Region). To confirm the NS1-Ag positive results and identify the serotype of the DENV, serotype-specific RT-PCR was performed using RNA directly isolated from the clinical serum samples. The combined positives from NS1-Ag and RT-PCR

have been used in this study. Out of 4045 RT-PCR performed, DENV-1 (n = 28), DENV-2 (n = 3702), DENV-3 (n = 45) and DENV-4 (n = 05) were identified. The remaining 265 were tested DENV-negative. Thus, the serotyping study indicates that the DENV-2 was the predominant serotype in circulation during the dengue outbreak in June to Nov 2023 in Delhi-NCR, India. As DENV-2 was identified as the predominant serotype (91.5%), with the objective of determining the genomic variability and potential genomic hotspots across the RT-PCR-confirmed DENV-2 population, whole genome sequencing of 3702 DENV-2 was performed using the ONT. Sequencing read analysis and mapping of the high-quality passed reads to a DENV-2 reference genome revealed differential genome coverage across the 3702 DENV-2 sequenced. Specifically, for a larger fraction (59.53%) of the DENV-2 population (n = 2189), 90–98.83% genome coverage was achieved. Moreover, for 434, the genome coverage was obtained in the range of 80–90%, whereas for 230, 167, and 235 DENV-2, 70–80%, 60–70%, and 50–60%, genome coverage was obtained, respectively (S2 Table). We were able to achieve genome coverage above 90 to 98.83% for a substantial number of the genomes. The differential genome coverage could be an attribute of the direct sequencing of the genomes from the clinical samples. It could also be a reflection of the virus copy number availability and integrity of the DENV's RNA genome in the clinical serum samples. At the same time, it is important to note the necessity of genomic surveillance with relevant clinical implications. Overall, out of 3702, 3254 DENV-2 have greater than 50% genome coverage. Given the importance of capturing genomic variability across a larger set of DENV genomes as it could have a possible implication in disease outcome or clinical manifestation across the patients, further analyses were conducted across the DENV-2 population and demonstrated greater than 50% genome coverage (n = 3254) (Fig 1A). As GISAID provides a platform for the genomic sequence dissemination in the global community, we took forward this cut-off along with the clinical implications mentioned above.

### Distinct Patterns of Synonymous and Nonsynonymous SNVs in DENV-2 genomes Linked with Viral Proliferation and Adaptation

Genetic variation may lead to a change in the properties and characteristics of the virus that could impact the epidemiology and associated virulence [33,34]. Therefore, we aim to investigate the variations in the genome of the DENV-2 population that caused an outbreak during June to November 2023 in Delhi-NCR, India. Variant analysis across the 3254 DENV-2 genomes yielded a total of 1618158 SNVs. All together, these SNVs were observed at a total of 3321 (out of 10723) genomic positions, representing 30.9% of the DENV-2 genome length. The total number of SNVs observed at a genomic position across our data set—the SNV spectrum—is represented by Fig 1B. As the variant analysis was conducted across the DENV-2 genomes with differential genome coverage, to ensure precise capturing of genomic variability, it is imperative to mention the SNV count across the DENV-2 genome vis-à-vis the genome coverage. Briefly, DENV-2 genomes with 90–98.83% coverage (n = 2189) exhibit SNV counts ranging from 437 to 741 per genome. Similarly, those with 80–<90% coverage (n = 434) show SNV counts between 471 and 689 per genome. Genomes with 70–<80% coverage (n = 230) display SNVs ranging from 345 to 575 per genome, while those with 60–70% coverage (n = 167) have SNV counts ranging from 310 to 531 per genome. Additionally, DENV-2 genomes with 50–<60% coverage (n = 235) exhibit SNV counts between 203 and 478 per genome (S8 Table).

Furthermore, to understand the specific impact of SNVs on DENV virus proliferation and pathogenesis, it is crucial to investigate the distribution of the SNVs across various genic regions. Therefore, we assessed the gene-wise load of the SNVs in our data set. Overall, the highest numbers of SNVs were observed across the NS5 region, with the 2699 bp long NS5 region recording 882 SNVs. Further, 525 SNVs were observed across the NS3 region. Moreover, across the 1484 bp long E-gene, a total of 470 SNVs were observed. To capture the relative abundance of the variations with respect to the gene length, we have determined the gene length-normalized count of SNVs across the various genic regions of DENV-2 populations. The results indicate relative overrepresentation of synonymous SNVs in prM, NS2A, NS2B, and NS4A (Fig 1C). On the other hand, the overrepresentation of non-synonymous SNVs was identified across the ancC, prM, and M proteins (Fig 1D).

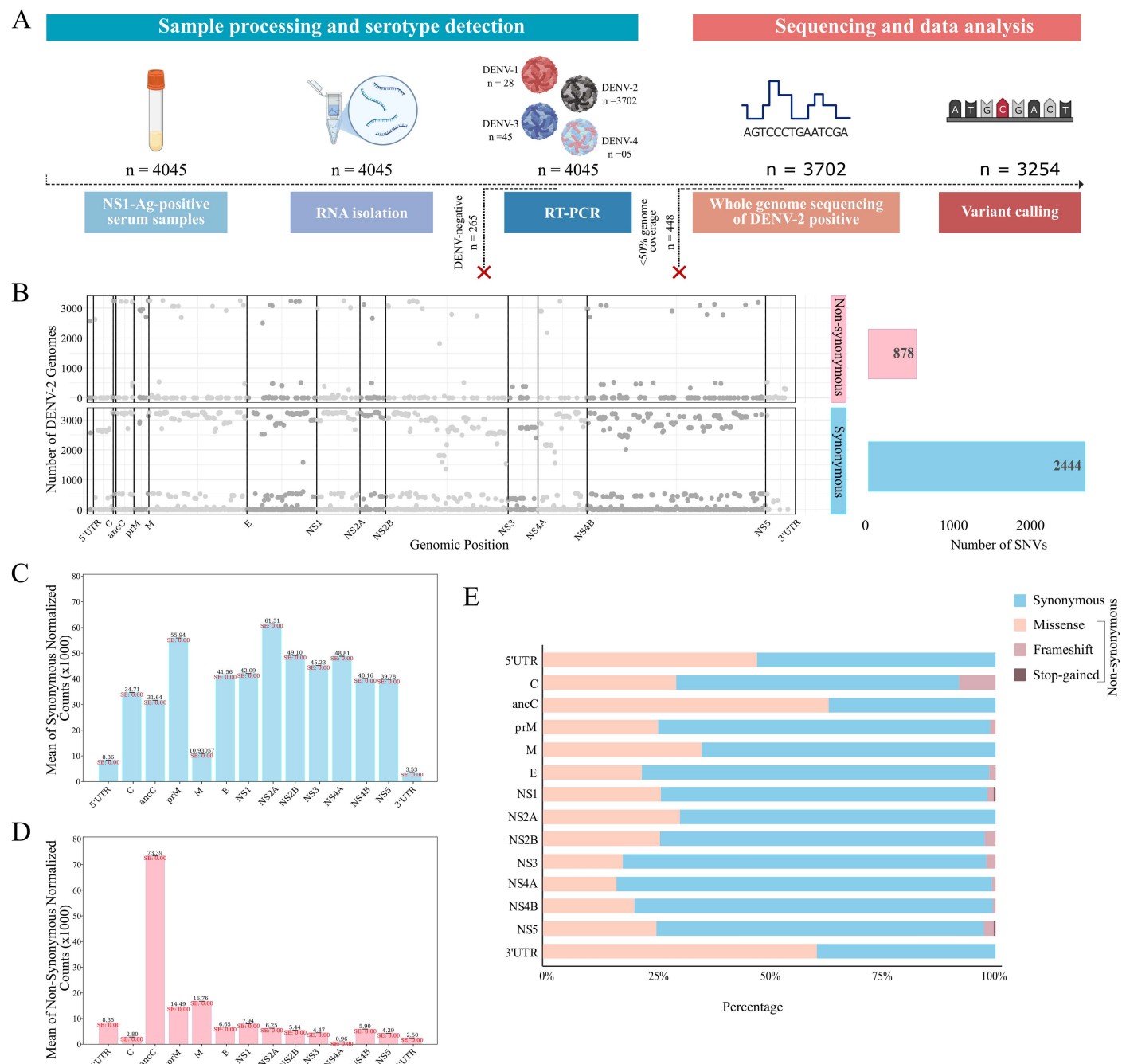

**Fig 1. Experimental workflow and SNVs spectrum across DENV-2 genomes. (A)** Study design includes NS1-Ag-positive serum sample collection, RNA isolation, and serotype detection, followed by whole genome sequencing, variant calling, and downstream data interpretation. This figure was created using licensed version of Biorender.com. **(B)** SNV spectrum with distribution of synonymous and non-synonymous SNVs across DENV-2 genomes whose coverage are > 50% (n = 3254). **(C and D)** Bar chart representing means of number of unique SNVs divided by gene length, a normalized count in the y-axis, with mean from bootstrap standard error estimation. **(C)** synonymous, and **(D)** non-synonymous SNV counts (x1000) across the DENV-2 genomes. **(E)** Horizontal stacked bar chart depicts the percentage proportion of synonymous and various types of non-synonymous SNVs—including missense, frameshift, and stop-gained—across different genes of DENV-2 genome.

Delving deeper into the SNV spectrum, out of the 3321 unique SNVs across the DENV-2 genomes, the SNVs observed at 878 genomic positions potentially produce variation in the amino acid sequence of the polyproteins that were designated as non-synonymous SNVs. The non-synonymous SNVs identified in our study are further classified into frameshift, missense, and stop-gained variation. The gene-wise proportion of each type of SNV, such as synonymous, frameshift, missense, and stop-gained, is depicted in Fig 1E. As evident from Fig 1E, the proportion of non-synonymous SNVs was comparatively high in structural proteins (ancC, M), 3'UTR, and 5' UTR, ensuing its role in viral entry, attachment, and replication of the viral genome. The proportion of synonymous SNVs appeared high in the E gene and non-structural genes, signifying its role in the adaptive evolution of the virus.

### Significant Alterations in the TLC, Platelets, and Liver Enzymes across DENV-2 Infected-Patients

The complete blood count (CBC) profiles of patients with dengue were utilized to observe the deviations in clinical parameters upon the onset of dengue infection. Clinical parameters were categorized as low, normal, or high based on the WHO-recommended reference ranges. S9 Table summarizes the distribution of these parameters across the defined categories. As it is evident from the S9 Table, several key clinical parameters were low across the dengue patients. Specifically, 51.96% (n = 889) dengue patients exhibited low total leucocyte count (TLC), indicating the development of leukopenia across the dengue patients. A total of 42.65% (n = 749) dengue patients were observed to have a low platelet count, suggesting thrombocytopenia. Additionally, other clinical parameters, such as haemoglobin, mean corpuscular volume (MCV), and mean corpuscular hemoglobin (MCH), were low in approx. 25% of the dengue patients, pointing to the development of anaemia associated with the DENV infection.

Moreover, the red cell distribution width (RDW) appears high in 52.75% (n = 901) patients, further supporting the diagnosis of anemia. Liver function test (LFT) data, available for a subset of patients, revealed elevated levels of SGOT-Aspartate Transaminase (AST) in 71.54% (n = 538) and SGPT-Alanine Transaminase (ALT) in 55.44% (n = 438) of dengue patients, markers often associated with severe dengue. Kruskal-Wallis with Dunn's comparison was performed to determine the significance of the deviations in clinical parameters from the normal range. Statistical analysis confirmed that deviations in clinical parameters across dengue patients were significant, underscoring the notable changes in blood parameters during dengue infection. At the same time, it highlights the need for evaluating whether differential clinical parameters are correlated with differential disease severities and potentially DENV-2 genomic SNVs within those patients.

### Leukopenia, thrombocytopenia and elevated liver enzymes differentiates dengue severity

As reported earlier [23–25], for analytical clarity we have been able to segregate 1294 dengue patients into mild (n = 473), moderate (n = 405), and clinically severe (n = 416) subgroups (Fig 2A), based on the available clinical data of the patients. Importantly, we observed a significant reduction in the TLC and platelet count in the clinically severe subgroup as compared to the patients from mild and moderate phenotype, highlighting a strong association of leukopenia and thrombocytopenia with increased disease severity (Fig 2B and 2C). Furthermore, a significant elevation in the hemoglobin levels was observed in clinically severe compared to the moderate sub phenotype, potentially reflecting hemo-concentration due to plasma leakage, a hallmark of severe dengue [35] (Fig 2D). In line with the previous findings [23], clinically severe subgroup exhibited a marked decrease in the neutrophil levels accompanied by an increase in the lymphocyte levels compared to the mild phenotype, further underscoring the dysregulated immune response in clinically severe dengue sub phenotype (Fig 2E and 2F). Several CBC parameters, including mean MCH, mean corpuscular hemoglobin concentration (MCHC), MCV, and mean platelet volume (MPV), were significantly elevated in the clinically severe phenotype compared to both mild and moderate subgroups, suggesting alterations in the erythropoietic and platelet homeostasis during severe disease [36]. However, no significant differences were observed in packed cell

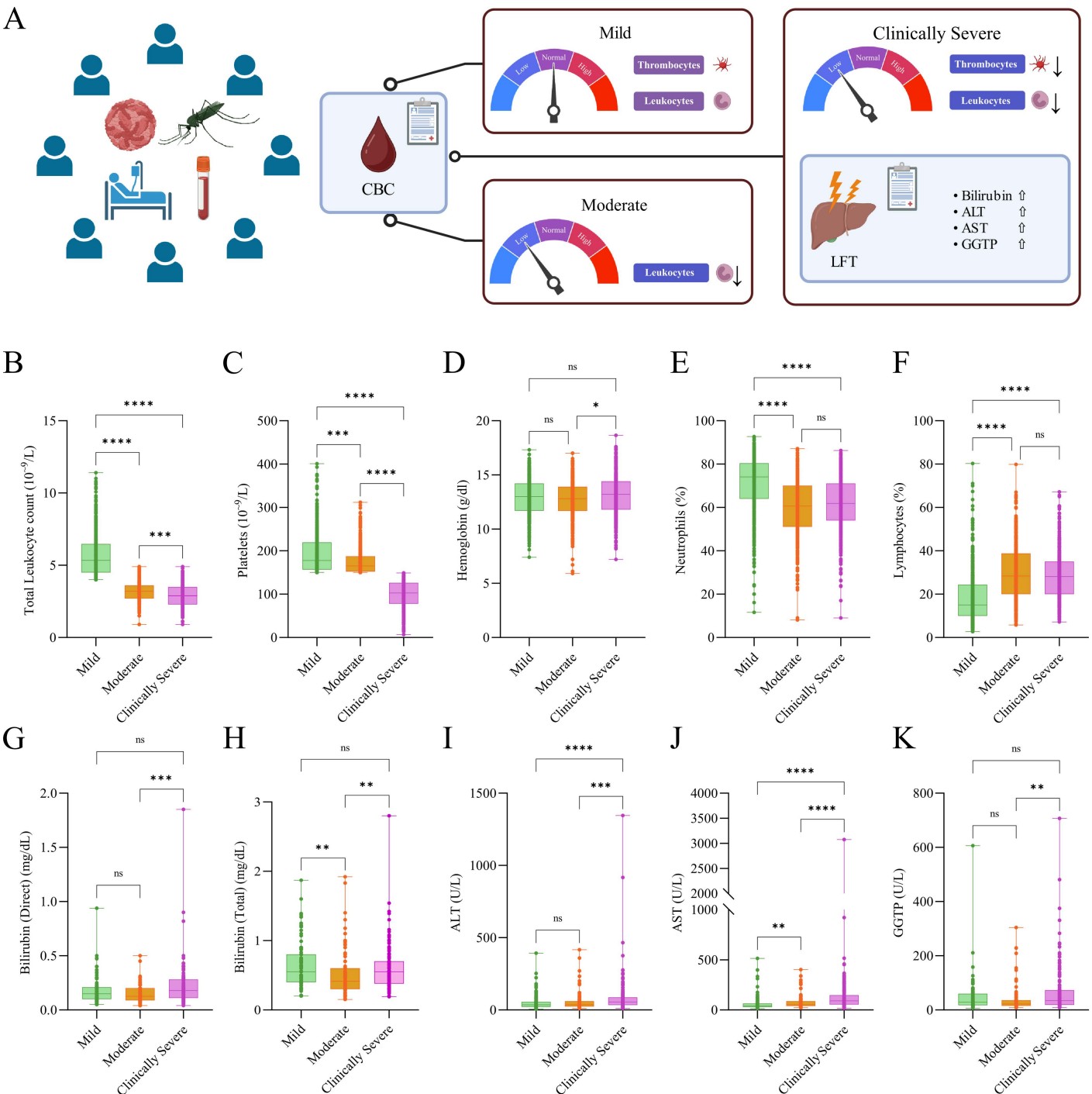

**Fig 2. Dengue severity subgrouping and comparative account of clinical manifestations. (A)** Schematic representation of the dengue patient categorization based on two primary CBC parameters: total leukocyte count (TLC) and platelet counts (thrombocytes) into mild, moderate, and clinically severe, and their association with fluctuations in CBC and LFT parameters during dengue virus infection. This figure was created using licensed version of Biorender.com. **(B-F)** Statistical comparison across the patient groups associated with mild, moderate, and clinically severe sub phenotypes for the major CBC parameters, including **(B)** total leukocyte count (10⁹/L); **(C)** platelets (10⁹/L); **(D)** hemoglobin (g/dl); **(E)** neutrophils (%); and **(F)** lymphocytes (%). **(G-K)** Statistical comparison across the patient groups associated with mild, moderate, and clinically severe dengue sub phenotypes for major LFT parameters, including **(G)** bilirubin (direct) (mg/dl); **(H)** bilirubin (total) (mg/dl); **(I)** ALT (U/L); **(J)** AST (U/L); and **(K)** GGTP (U/L). Kruskal-Wallis with

Dunn's comparison was performed for comparison of each clinical parameter across mild, moderate, and clinically severe. The significance value is denoted as *, where * indicates p ≤ 0.05, ** indicates p ≤ 0.01, *** indicates p ≤ 0.001, and **** indicates p ≤ 0.0001.

volume (PCV), red blood cell count (RBC), or red cell distribution width (RDW) across the severity subgroups, indicating a degree of stability in these parameters despite disease progression (S1 Fig).

Previous studies have highlighted deviations in liver function parameters during dengue virus infection, which often correlate with the disease severity and systemic inflammation [9,37,38]. Consistent with this, the liver function test (LFT) revealed significant alterations in the liver profile of patients associated with clinically severe phenotype compared to the mild and moderate phenotypes. Clinically severe subgroup showed marked increase in bilirubin (both total and direct), SGOT-Aspartate Transaminase (AST), SGPT-Alanine Transaminase (ALT), and GGTP (Gamma GT) serum compared to mild and moderate sub phenotypes, indicating hepatic dysfunction potentially driven by the immune-mediated liver injury or direct viral effects (Fig 2G-2K). However, no significant differences were observed in the albumin-to-globulin (A/G) ratio or alkaline phosphatase levels across the severity subgroups, suggesting these parameters might remain unaffected by the disease progression (S1 Fig). Together, these findings reinforce the critical role of haematological and hepatic parameters in delineating the pathophysiology of the patients associated with clinically severe phenotype.

## Monophyletic clustering of cosmopolitan DENV-2 genotypes from India

DENV-2 is classified into six genotypes: Asian I, Asian II, American, American-Asian, Cosmopolitan, and Sylvatic, with the Cosmopolitan genotype being the most prevalent in India [39]. To determine the genotype of the DENV-2 strains in our study, we conducted genotype identification and observed that DENV-2 genomes in our study cohort exclusively belonged to the Cosmopolitan genotype.

Further, a comprehensive phylogenetic analysis was performed to investigate the relationship of the in-house sequenced DENV-2 genomes with the global strains. For this, 7,976 DENV-2 genomes with >90% coverage were retrieved from GISAID, representing diverse geographical regions, including India, Asia, Africa, North America, and South America. The analysis revealed distinct phylogenetic clusters reflecting global diversity in the DENV-2 strains (**Fig 3**). Our in-house genomes clustered as a separate Clade indicating the independent evolutionary pattern of the DENV-2 population from India. Further, it appears that the DENV-2 genomes captured from the outbreak in India were monophyletic and uniform cosmopolitan genotypes, thus providing a more rational DENV-2 population for genetic variability analysis. We further investigated potential clustering of the DENV-2 genomes across the severity groupings, however no possible clusters were identified (**Fig 4**) These findings highlight the distinct genetic signatures of the in-house DENV-2 strains and their divergence from globally circulating genomes.

## Distinct genomic hotspots in E, NS4B and NS5 regions underpin dengue disease severity

Aligning with the theme of our study, we investigated the possible association of SNVs observed across the DENV-2 genomes and the clinical parameters of the dengue patients. As mentioned in the study design, the SNVs association study was conducted across 1,294 patients with detailed clinical data. The analytical workflow for the SNVs association study is depicted in Fig 5A, whereas Fig 5B explains the genome coverage vis-à-vis the SNV count in 1,294 DENV-2 genomes. Across the 1,294 DENV-2 genomes, SNVs were identified at 2,496 genomic positions. Many of these SNVs were commonly distributed across the mild, moderate, and clinically severe subgroups, while several SNVs were uniquely distributed in each subgroup, underscoring potentially distinct genomic hotspots/signatures linked to disease severity (Fig 5C). Specifically, the mild subgroup exhibited 485 unique SNVs not observed in the moderate or clinically severe patients. Similarly, the moderate group had 333 unique SNVs, while the phenotype associated with clinically severe subgroup demonstrated 308 exclusive SNVs (Fig 5C). The frequency of the SNVs associated with mild, moderate, and clinically severe phenotypes are depicted in Fig 5D.

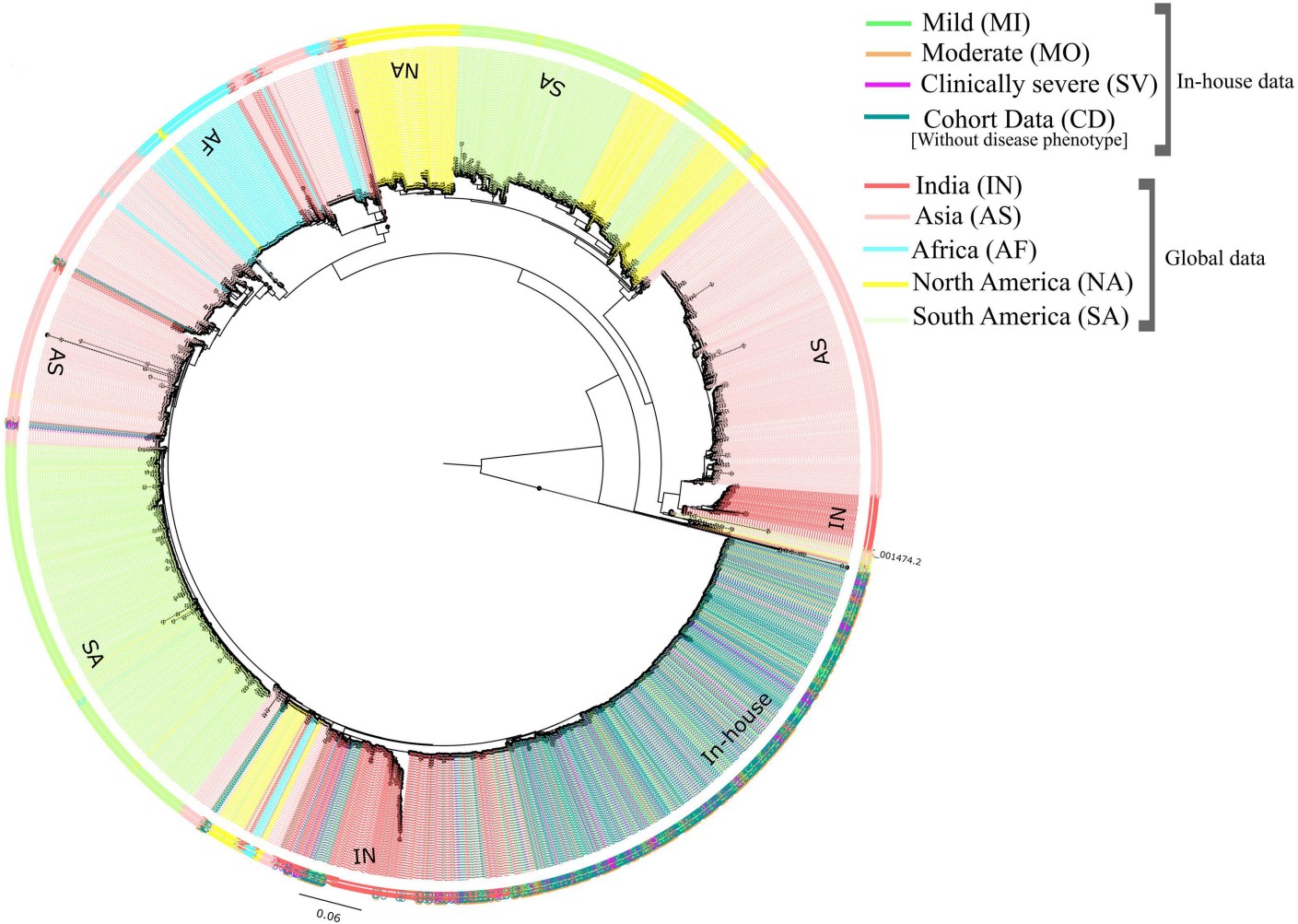

**Fig 3. Phylogenetic analysis of 10164 DENV-2 genome sequences.** The distribution of genomes of different continents from various outbreaks, along with the in-house sequences from a single outbreak, is represented using distinct colors. The color codes are as follows: mild (green), moderate (orange), clinically severe (magenta), cohort data (sea green), India (salmon red), Asia (pink), Africa (cyan), North America (yellow), and South America (yellow-green).

Further, we aim to explore the involvement of SNVs in the clinical manifestation of the DENV-2 infected patients. For this analysis, the presence and absence of SNVs in the DENV-2 genes across the phenotypes associated with mild, moderate, and clinically severe groups was tabulated, and a chi-square test was applied (S5 Table). We observed the significant SNV burden across the *E*, *NS2A*, *NS4B*, *NS5*, and *3'UTR* for the mild vs. clinically severe comparison, indicating their role in differential severity, whereas for the mild vs. moderate group comparison, *5'UTR*, *M*, *E*, *NS1*, *NS2A*, *NS4B*, and *NS5* were observed having the significant burden of SNVs. However, only *3'UTR* was noted for the presence of SNV burden in moderate vs. clinically severe sub phenotypes. To further investigate the significance and potential functional relevance of SNVs distributed across these significant genes, fisher's exact test and phi-coefficient correlation analyses were performed. Our analysis identifies statistically significant SNVs associated with each severity category. Comparisons between mild and moderate subgroup of patients revealed 14 significant SNVs that were uniquely associated with the mild patients (Fig 5E). The mild vs. clinically severe group analysis yielded 83 significant SNVs, of which 56 were associated

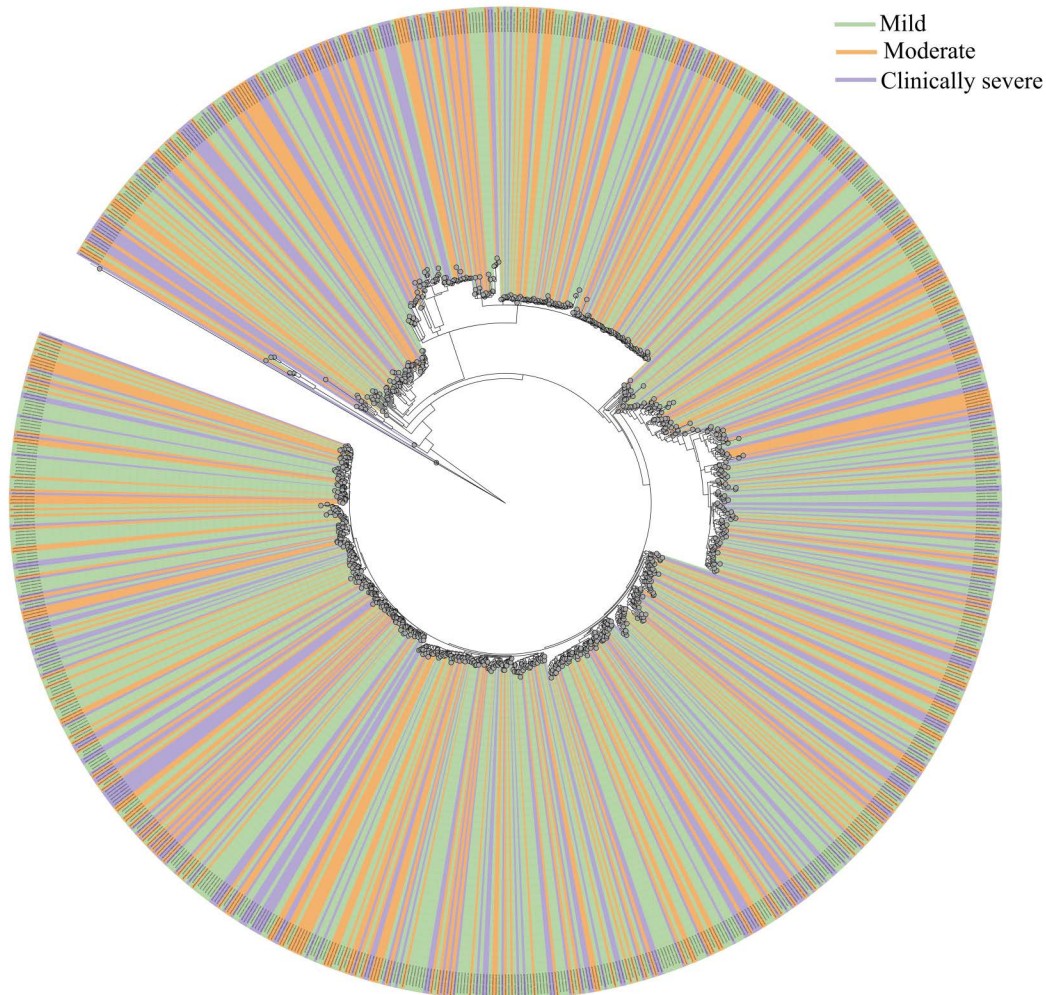

**Fig 4. Phylogenetic analysis for the DENV-2 genomes associated with different disease sub-phenotypes.** A magnified view of phylogenetic tree for 929 genomes comprising clinical sub-phenotypes mild (light green), moderate (orange), and clinically severe (magenta).

with mild cases and 27 with clinically severe patients (Fig 5E). However, no SNVs were found significant between moderate and clinically severe patients. Interestingly, the 14 SNVs significantly associated with mild group across mild vs. moderate, were also observed in the mild vs. clinically severe group, indicating exclusive association of these SNVs with mild phenotypes. These results provide evidence of distinct SNV patterns correlating with disease severity.

To assess the functionality and potential biological significance of statistically significant SNVs across the severity subgroups, we looked into the distribution of these SNVs across the DENV-2 genome. This analysis revealed intriguing clustering patterns of SNVs within the genic regions that differed between the severity subgroups. Fig 6 demonstrates the distribution of significant SNVs associated with each severity subgroup. For the mild vs. moderate group comparison, all significant SNVs associated were predominantly clustered in the *NS4B* regions in the mild patients. However, we did not observe any significant SNVs associated with the moderate group for the same comparison. For the moderate vs. clinically severe group analysis, none of the SNVs were observed to be significantly associated with any of the phenotypes. Across the comparison of mild vs. clinically severe phenotypes, 56 significant SNVs associated with mild group

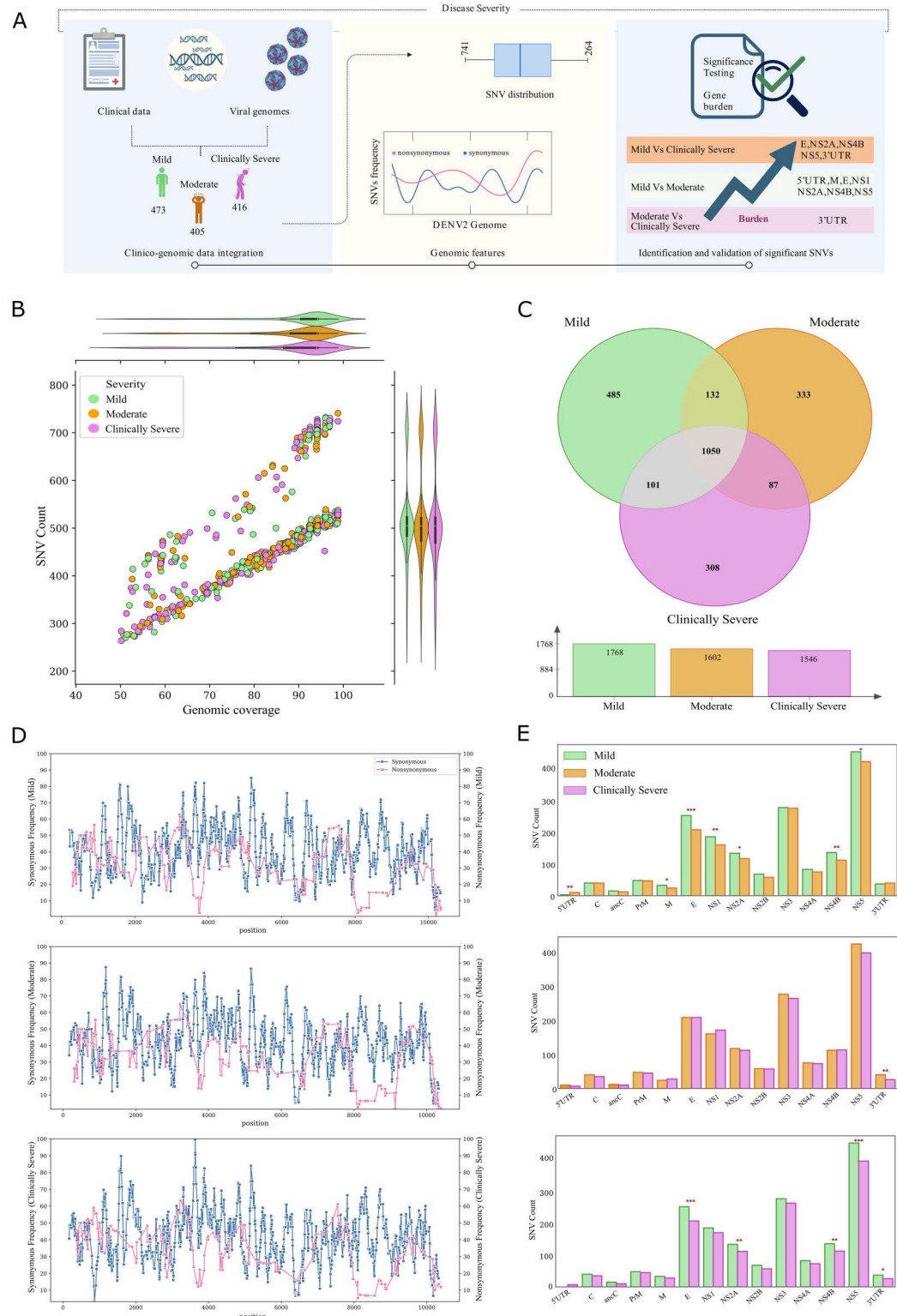

**Fig 5. Association of SNVs with differential disease severity. (A)** The analytical workflow of the SNV association studies, showing a representation of clinico-genomic integration of mild, moderate, and clinically severe subgroups with differential genomic features, that is depicted through differential SNV burden in the genes for different categories. This figure was created using licensed version of Biorender.com. **(B)** Joint-grid plot depicting genome

coverage and SNVs count across mild, moderate, and clinically severe sub phenotypes depicting a positive correlation for higher number of SNVs for higher coverage. **(C)** Venn diagram demonstrating the unique and common SNVs across the mild, moderate, and clinically severe patients. **(D)** Frequency of SNVs in mild, moderate, and clinically severe groups across the DENV-2 genomes has been smoothed by moving average in intervals of 10, with the average represented by square markers. **(E)** Bar plot depicting significant burden of SNVs across the gene of DENV-2, with the significance shown in the asterisk symbol obtained from the chi-square test of independence.

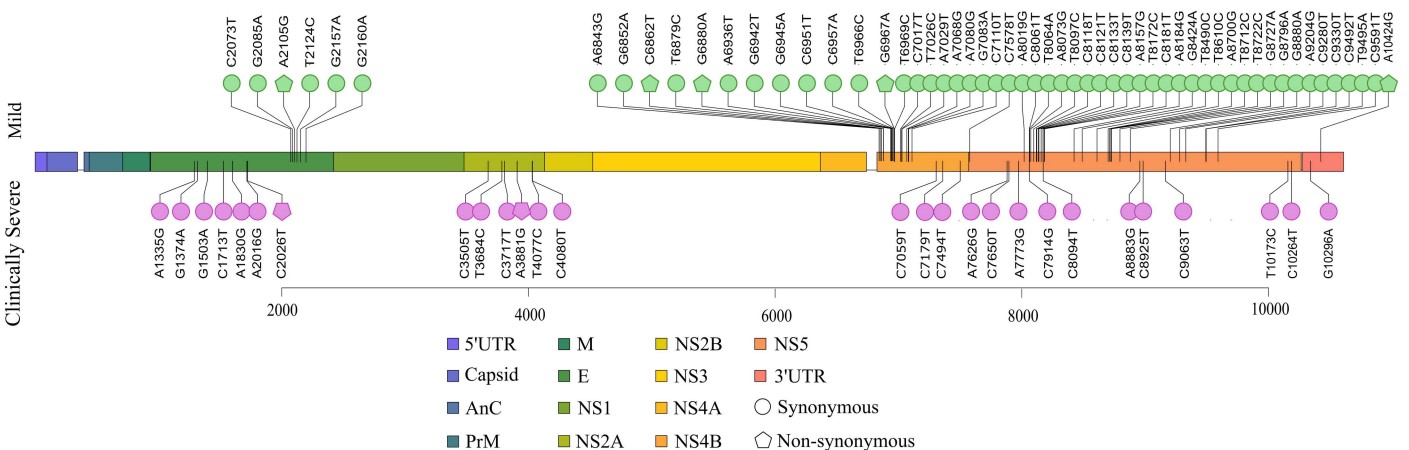

**Fig 6. Distribution of significant SNVs across disease severity.** Lollipop plot depicting significant SNVs across mild and clinically severe phenotypes. SNV is represented with different colors, signifying individual groups. Significance was calculated using fisher's exact test (p < 0.05). The directionality and strength of association of significant SNVs with respective groups were confirmed with the phi coefficient correlation test.

were distributed across *E*, *NS4B*, *NS5*, and *3'UTR* regions, with the majority of the SNVs cluttered in the *NS4B* and *NS5* regions. Interestingly, of these 56 SNVs associated with mild group across the mild vs. clinically severe comparison, 14 were common for the mild vs. moderate group and were exclusively present in the *NS4B* region. For the mild vs. clinically severe comparison, the clinically severe patients revealed unique associations of distinct SNVs distributed across the *E*, *NS2A*, *NS4B*, *NS5*, and *3'UTR* regions, with the *E* and *NS5* genes having their predominant contribution of SNVs towards the clinically severe phenotype.

Our domain-wise analysis of SNVs in the E-gene reveals a clear demarcation in the distribution of SNVs associated with clinically severe and mild phenotype, highlighting their potential functional significance in disease severity. SNVs associated with clinically severe phenotype predominantly occur in the DI-DIII region [A1335G, G1374A, G1503A, C1713T, A1830G, A2016G, and C2026T], which is critical for receptor binding, membrane fusion, and viral entry [40]. In contrast, SNVs associated with mild sub-phenotype are localized in the DIII and stem regions [C2073T, G2085A, A2105G, T2124C, G2157A, and G2160A], which are primarily involved in the virus assembly and host cell recognition [41]. The non-synonymous SNVs C2073T and A2105G result in amino acid changes in the viral proteins, while synonymous SNVs are likely to affect viral processes through codon usage bias, translation efficiency, transcript stability and interactions with host factors. Notably, a study by Luciana et al. [42] highlights the selection of synonymous mutations that favor DENV's preferred nonoptimal codons, which enhance viral fitness. Further targeted studies in this direction would further strengthen these potential functional understanding of the synonymous mutations. Additionally, in the NS4B and NS5 regions, we observe a mixture of SNV associations, suggesting diverse functional impacts on viral replication, immune modulation, and overall disease progression. Notably, the non-synonymous SNVs modulate function through alteration in the amino acid sequence of the protein and could have detrimental effects for the host. Therefore, we specifically looked further into the non-synonymous SNVs and their possible association with the dengue severity across different disease

comparison subgroups. We identified a total of 5 non-synonymous SNVs (E: Asn390Ser; NS4B: Leu13Phe, Ala19Thr, Val48Ile and 3'UTR: X3475Trp) associated with mild phenotype. Further, two non-synonymous SNVs (E: Pro364Ser and NS2A: Lys135Arg) were observed to be associated with clinically severe phenotype. Altogether, these findings underscore the distinct genomic signatures associated with dengue severity, highlighting the E, NS4B, and NS5 regions as critical hotspots for SNVs in dengue differential severity.

### Statistically significant SNVs showing global frequency flip for mild and clinically severe patients

To understand the occurrence and frequency of the significant SNVs associated with the disease severity across the globe, we have conducted SNVs analysis and frequency comparison across the genomes of DENV-2 reported worldwide. Then we looked for the statistically significant SNVs across the global dataset. We observed the global occurrence of these SNVs with varying frequencies in the country cohort across the globe. The frequency comparison of the statistically significant SNVs with the global SNVs was represented through a matrix plot (Fig 7). Interestingly, the significant SNVs associated with mild phenotype are having uniform frequency in the minimum of two countries, while clinically severe phenotype have frequency flip in all the SNVs.

### Discussion

Dengue is an epidemic disease that, in recent years, has been rising alarmingly and transgressing into newer regions globally—primarily due to evolving SNV rates—thus causing a major concern worldwide and emphasizing the need for clinical and community genomic surveillance on a larger scale to evaluate the evolutionary changes within the dengue virus genome and elucidate the clinical pathophysiology. The inherent genomic plasticity of DENV, primarily attributed to low-fidelity RdRp, induces genetic variation and allows the virus to rapidly adapt to host immune pressures, often leading to significant alterations in its pathogenicity and fitness [43]. More importantly, a growing body of evidence also suggests the role of genetic variability in the pathogenesis of dengue [19,44,45]. Surprisingly, the pathogenesis of dengue with differential severity is poorly understood vis-à-vis genomic SNVs. The major reason could be the lack of a substantial number of studies focusing on exploring the role of genetic variability in pathogenesis. Prior to this investigation, very few studies had been conducted in this direction. Particularly, a study reports the genome sequence of five DENV-2 and captures the amino acid diversity [46]. Further, a study conducted in Rajasthan, India, utilises a single DENV-3 genome for

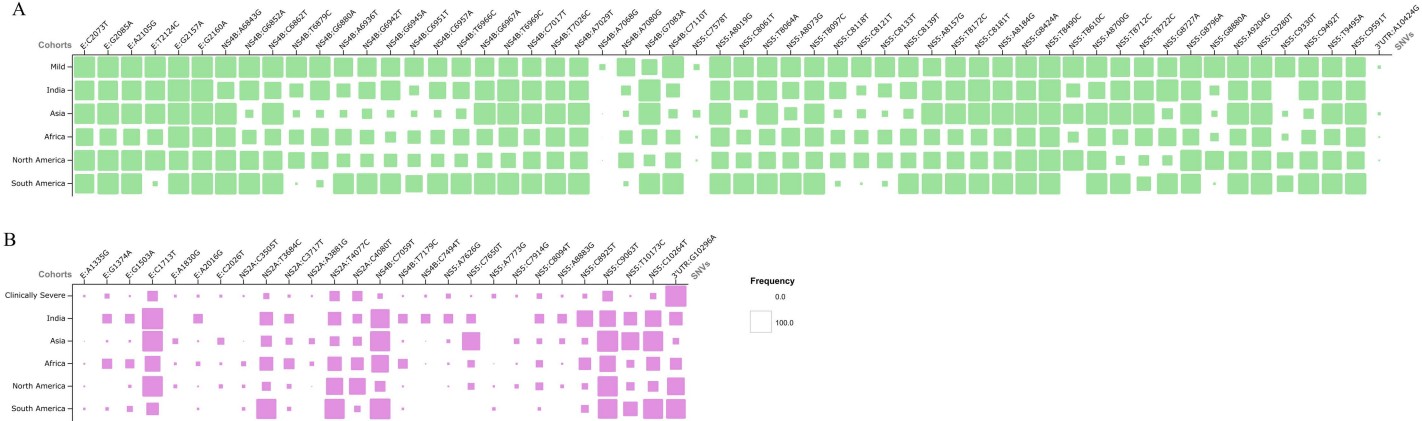

**Fig 7. Comparison of statistically significant SNVs associated with mild and clinically severe phenotype with global frequency (A-B).** The matrix plot depicts the comparison of significant SNVs identified in the of DENV-2 genomes from the study with the global frequencies reported worldwide A) Mild, and B) Clinically severe.

capturing genetic variability. Even with studies conducted across the globe, the genetic variability captured in dengue virus genomes needs to extend its scale and number. With DENV-2 identified as the predominant serotype in circulation during the 2023 outbreak in New Delhi, India, we have utilized it to capture the genomic variability across the genome. A key novelty of the present study lies in the scale and dissemination of a substantial number of DENV-2 genomes from a single outbreak that happened in an endemic region. Moreover, our study reveals significant insights into the SNVs landscape of the DENV-2 population, highlighting the extent of genetic variability. Unravelling the spectrum of SNVs across the DENV-2 genomes at this scale would pave the path for understanding and determining its role in the pathogenicity of dengue with differential severity.

The pathogenicity of dengue disease could be an intricate attribute of the DENV gene expression and functioning of gene products. Therefore, the observation of SNVs across the DENV gene with the potential to change the transcript and/or lead to amino acid variation in the polyproteins could have an implicit effect on dengue pathogenicity and severity. Our study showed the predominance of synonymous SNVs in non-structural genes of DENV-2 that could potentially modulate the functions through changes in transcript and play a role in the facilitation of an adaptive evolutionary strategy for the viruses [47]. Additionally, a comprehensive catalogue of nonsynonymous SNVs identified in structural proteins such as ancC and M, as well as in the 3' and 5' UTRs, regions indicates their potential role in the clinical outcome through modulating the viral entry, attachment, replication, and immune evasion, underscoring their involvement in the viral pathogenesis [47].

Following the capture of the genomic variability across the DENV-2 population, we have comprehensively analysed the clinical manifestations across the dengue patients to facilitate the integrative analysis of genomic variability vis-à-vis clinical parameters. Our analysis across clinical data sets identifies leukopenia and thrombocytopenia as major clinical features of the dengue patients. It is interesting to note that thrombocytopenia often leads to hemorrhagic manifestation in dengue infection, leading to the severe disease [48]. The observation of leukopenia across the dengue patients could be the manifestation of immune-suppression due to direct viral effects on bone marrow or immune-mediated destruction of white blood cells [48]. Moreover, thrombocytopenia and leukopenia could be an outcome of the DENV infection- mediated immune system's inability to control viral replication [23,49]. Importantly, the significantly more pronounced thrombocytopenia and leukopenia in clinically severe cases indicate its implications as an early indicator of dengue severity prediction and emphasizes the need of inferring the TLC and platelet count from the clinical data set of the dengue patients' vis-a-vis the SNVs profile of the virus that would potentially pave the path for understanding the progression of infection towards varying level of disease severity. Additionally, the significant elevation in haemoglobin levels in clinically severe cases suggests plasma leakage-induced hemo-concentration—a defining characteristic of severe dengue [35]. The increase in hemoglobin across the dengue patients could be an attribute of fluid loss from the vascular compartment into interstitial spaces that might be contributing to shock and other complications related to severe dengue [36]. Elevated red cell distribution width (RDW) in 52.75% of patients further supports anaemia as a key haematological feature, potentially arising from erythropoietic dysregulation triggered by the DENV infection [50]. High RDW is often associated with inflammation and oxidative stress, contributing to impaired red blood cell production and turnover in severe dengue [50]. Clinically severe cases, however, exhibited marked deviations in MCH, MCHC, MCV, and MPV. These changes likely reflect erythropoietic and platelet homeostasis alterations associated with clinically severe disease, aligning with previous studies reporting similar haematological shifts during severe dengue [25]. These findings underscore the potential of these parameters as indicators of disease severity and the systemic impact of dengue infection.

In addition to the CBC parameter, the hepatic dysfunction has also been documented in severe dengue cases [9,38]. In our study, a subset of patients depicting both leukopenia and thrombocytopenia had also demonstrated elevated levels of bilirubin, SGPT, and GGTP—key markers of liver damage—indicate extensive hepatic damage/dysfunction, likely mediated by a combination of direct viral cytotoxicity and immune-driven injury [14]. Furthermore, two key liver enzyme levels, ALT and AST, were also observed to be high from the normal range, corroborating the disease severity of the dengue

patients in our study cohort. Especially, the liver enzyme, AST level, was observed significantly elevated in the dengue patient's subgroup exhibiting thrombocytopenia as well as those with both thrombocytopenia and leukopenia compared to reference group (no thrombocytopenia and no leukopenia), suggesting that AST level in dengue patients could potentially be used as the prognostic marker for both thrombocytopenia and leukopenia (S10 Table). While their clinical utility is evident, these findings also emphasize the need for a more nuanced understanding of how the dengue virus affects different physiological systems.

The genetic variability is known to affect viral virulence and modulate clinical outcomes, emphasizing the need to evaluate its role in the clinical manifestations amongst the dengue patients [4,18,19,44,45]. Therefore, we employed a clinico-genomic approach and conducted an integrative analysis of genetic variability across a data set comprising 3254 in-house sequenced DENV-2 genomes vis-à-vis clinical parameters of dengue patients. Our analysis revealed distinct patterns with unique sets of SNVs associated with each severity group, thus offering insights into potential genomic signatures associated with disease severity in dengue virus infection. The predominance of high-frequency SNVs associated with mild and moderate cases suggests these SNVs may represent adaptive changes that optimize viral fitness without inducing excessive host damage, potentially facilitating more efficient transmission [51]. In contrast, low-frequency unique SNVs observed in clinically severe cases potentially modulate the critical host-pathogen interactions through their individual effects, driving severe pathogenesis [52]. Earlier research related to viral evolution highlights that rare SNVs can enhance virulence or facilitate immune evasion but may come at the cost of reduced replication efficiency or fitness in non-optimal environments [53,54]. The observation of a unique set of low-frequency SNVs in clinically severe cases and high-frequency SNVs in mild and moderate cases in our study cohort aligns with broader concepts in viral evolution, where rare SNVs often lead to immune escape or increased virulence, as observed in other RNA viruses, but they frequently impose fitness costs that hinder their persistence in the population [55].

Statistical analyses further identified significant SNVs associated with severity subgroups, reinforcing the notion of distinct SNV signatures. In mild cases, significant SNVs were enriched in the NS4B region, SNVs in these regions may modulate viral replication dynamics without heavily perturbing host cellular processes, thus contributing to milder disease manifestations [56]. In contrast, SNVs associated with clinically severe patients exhibited a more dispersed distribution across structural and non-structural regions with a dominance of SNVs in the E and NS5 regions. The clustering of clinically severe-associated SNVs in the E and NS5 regions points to their potential impact on viral assembly and entry mechanisms, which are critical for infectivity [18]. Moreover, the clinically severe associated SNVs in NS5 are particularly noteworthy, given their established roles in immune evasion and replication fidelity [57,58]. Similarly, SNVs in NS5, which encodes the viral RNA-dependent RNA polymerase [59], could alter replication fidelity or interfere with host immune signalling pathways [60], exacerbating disease severity.

Prior to this study, few investigations have linked SNVs in the E and NS5 regions of DENV-2 to disease severity. For example, an isoleucine-to-valine substitution at position 322 in EDIII of the E gene has been associated with both dengue fever (DF) and dengue hemorrhagic fever (DHF) [19]. Similarly, the Gly605Val mutation in NS5 has been linked to increased virulence and severe dengue outcomes [21], while T7812G (Gly81Gly) and C9420A (Ala617Ala) substitutions are thought to enhance viral replication [57]. While SNVs in the 3'UTR and their epidemiological roles are well studied, our findings highlight distinct evolutionary pressures in dengue virus adaptation. Specifically, the statistically significant SNV from 3'UTR, A10424G associated with mild phenotype belong to fNR2 (flaviviral nuclease-resistant RNA) region, could potentially contribute to maintaining sfRNA (subgenomic flavivirus RNA) stability which may play a role in efficient transmission [61]. Conversely, other significant SNV (G10296A) associated with clinically severe phenotype in the sfRNA region, may potentially enhance the immune evasion properties, since the sfRNA region primarily involves interactions with the host factors like TRIM25 [62]. This suggests a potential balance between viral fitness and immune adaptation, influencing disease severity. However, further evidence is required to confirm these potential functional roles of the SNVs identified in our study using clinical patients.

More importantly, the distribution of significant SNVs associated with disease severity in the specific gene of DENV-2 would help us in understanding the mechanism of virus-virulence. Additionally, the identification of genomic regions hosting SNVs significantly associated with mild and clinically severe phenotypes highlight its implication to be explored as a potential therapeutic target in the virus genome that could be used to control and devise the early mitigation strategy towards the progression of DENV infection severity. Furthermore, the correlation of significantly associated SNVs with the key clinical features including thrombocytopenia and leukopenia emphasizes the need of extensive clinico-genomic surveillance to unravel the dengue disease severity.

To investigate the occurrence of severity-associated SNVs in the global DENV-2 population, we conducted variant analysis on 8085 DENV-2 genomes from the GISAID database. Our analysis revealed these SNVs in a global cohort with varying frequency. It's worth noting that these particular SNVs had not been documented before. The identification of previously unreported severity-associated SNVs in global DENV-2 genomes further highlights the importance of comprehensive genomic surveillance, especially in the endemic regions to capture viral evolution (Fig 8).

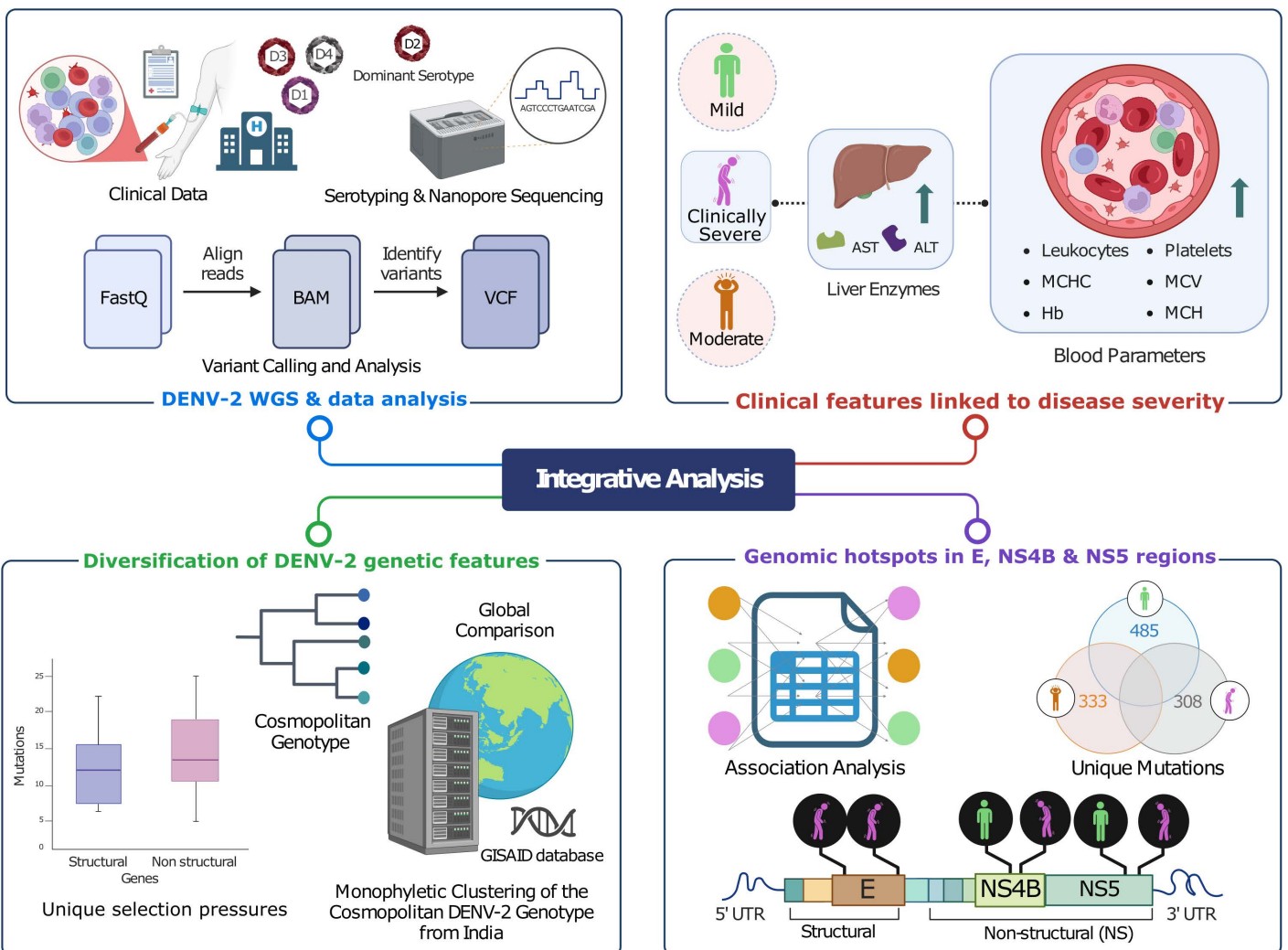

**Fig 8. Summary of the identification of SNV hotspots in the DENV-2 genome.** It highlights E, NS4B and NS5 regions in the mild and clinically severe dengue sub-phenotypes through an integrative clinico-genomic approach. This figure was created using licensed version of Biorender.com.

The discovery of these variants across our data set highlights the potential for improved understanding of dengue disease progression and severity locally as well as across the globe. These results open up opportunities for more focused studies, potentially improving the prediction and treatment of severe dengue cases worldwide.

## Limitations

While our study provides significant insights into the genomic variability of DENV-2 and its potential association with disease severity sub-phenotypes, few limitations must be acknowledged for future consideration/s. First, the absence of key clinical markers such as cytokine profiling and immune response indicators limits a more comprehensive understanding of host-pathogen interactions. The cross-sectional nature of our study also restricts insights into the temporal evolution of SNVs, highlighting the need for longitudinal analyses across multiple outbreaks. Moreover, as our dataset is derived from a single outbreak in a specific geographic region, future studies incorporating diverse viral strains from multiple endemic regions are essential for broader applicability of these findings. Potential biases in sequencing representation must also be considered, as our dataset primarily includes symptomatic cases, and the inclusion of asymptomatic infections could offer a more complete perspective on viral adaptation and transmission dynamics. Lastly, while our study identifies rare SNVs that may contribute to immune evasion or increased virulence, their precise mechanistic roles in dengue pathogenesis require further research. Addressing these limitations in future studies will provide a more comprehensive understanding of DENV-2 evolution, its clinical implications, and potential mechanisms driving disease severity.

## Supporting information

**S1 Table. List of PCR primer sequences used for DENV-2 genome amplification.**
(XLSX)

**S2 Table. Ct-value, Read Count, Coverage Depth and Breadth for 3254 DENV-2 genomes.**
(XLSX)

**S3 Table. Annotation of SNVs captured across 3254 DENV-2 genomes.**
(XLSX)

**S4 Table. Details of SNVs across 1294 DENV-2 genomes and their distribution in severity subgroups.**
(XLSX)

**S5 Table. Significant SNV burden across DENV-2 genes.**
(XLSX)

**S6 Table. Fisher Exact and Phi-Correlation Co-efficient for significant SNVs.**
(XLSX)

**S7 Table. Details of global DENV-2 genomes (n = 10164) included in the phylogenetic study.**
(XLSX)

**S8 Table. DENV-2 Genome Coverage vis-a-vis SNVs count.**
(XLSX)

**S9 Table. Significant deviations in clinical parameters of dengue-infected patients (Kruskal-Wallis with Dunn's comparison).**
(XLSX)

**S10 Table. Comparison of clinical parameters between the reference group, thrombocytopenia, leukopenia and combined thrombocytopenia-leukopenia.**
(XLSX)

**S1 Fig. Statistical analysis of CBC and LFT parameters across mild, moderate, and clinically severe patients.**
(PDF)

AcknowledgmentThe authors duly acknowledge all the dengue patients who participated in the study. Authors acknowledge the help and support from Dr. Bharti Kumari towards facilitation as a project manager and Dr. Aradhita Baral for coordination with the funders. The authors acknowledge the support of Anil Kumar and Nisha Rawat towards dengue sample transport and sample management. KK, AY, KC and N., acknowledge Council of Scientific & Industrial Research (CSIR), India for the fellowship support. JS acknowledges the University Grants Commission (UGC) for the research fellowship.

## Author contributions

**Conceptualization:** Md Imran, Rajesh Pandey.

**Data curation:** Varsha Ravi, Kriti Khare.

**Formal analysis:** Varsha Ravi, Kriti Khare, Pallavi Mishra.

**Funding acquisition:** Rajesh Pandey.

**Investigation:** Varsha Ravi, Kriti Khare, Ramakant Mohite, Sayanti Halder, Richa Shukla, Chinky Shiu Chen Liu, Aanchal Yadav, Jyoti Soni, Kanika, Md Imran, Rajesh Pandey.

**Methodology:** Varsha Ravi, Kriti Khare, Md Imran.

**Project administration:** Rajesh Pandey.

**Resources:** Komal Chaudhary, Neha, Bansidhar Tarai, Sandeep Budhiraja, Pooja Khosla, Tavpritesh Sethi, Rajesh Pandey.

**Software:** Varsha Ravi, Pallavi Mishra.

**Supervision:** Rajesh Pandey.

**Validation:** Varsha Ravi, Md Imran, Rajesh Pandey.

**Visualization:** Varsha Ravi, Kriti Khare, Ramakant Mohite, Md Imran.

**Writing – original draft:** Varsha Ravi, Kriti Khare, Ramakant Mohite, Md Imran.

**Writing – review & editing:** Rajesh Pandey.

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
