## [Decision Letter · Decision Letter 0]

29 Jan 2025

PNTD-D-24-01925

Genomic hotspots in the DENV-2 serotype (E, NS4B, and NS5 genes) are associated with Dengue Disease Severity in the Endemic Region of India

Dear Dr. Pandey,

Thank you for submitting your manuscript to PLOS Neglected Tropical Diseases. After careful consideration, we feel that it has merit but does not fully meet PLOS Neglected Tropical Diseases's publication criteria as it currently stands. Therefore, we invite you to submit a revised version of the manuscript that addresses the points raised during the review process.

Please submit your revised manuscript within 60 days Mar 30 2025 11:59PM. If you will need more time than this to complete your revisions, please reply to this message or contact the journal office at plosntds@plos.org. Please include the following items when submitting your revised manuscript:

We look forward to receiving your revised manuscript.

Kind regards,

Zach N Adelman

Guest Editor

David Safronetz

Section Editor

Shaden Kamhawi

co-Editor-in-Chief

Paul Brindley

co-Editor-in-Chief

**Additional Editor Comments:**

Please address the concerns noted by the reviewers. In particular, multiple reviewers made concerns regarding the classifications of mild, moderate and severe dengue and be sure all requested adjustments to the methods/sequence analysis are incorporated.

**Journal Requirements:**

At this stage, the following Authors/Authors require contributions: Varsha Ravi, Kriti Khare, Ramakant Mohite, Pallavi Mishra, Sayanti Halder, Richa Shukla, Chinky Shiu Chen Liu, Aanchal Yadav, Jyoti Soni, Kanika Kanika, Komal Chaudhary, Neha Neha, Bansidhar Tarai, Sandeep Budhiraja, Pooja Khosla, Tavpritesh Sethi, Md Imran, and Rajesh Kumar Pandey. Please ensure that the full contributions of each author are acknowledged in the "Add/Edit/Remove Authors" section of our submission form.

2) Some material included in your submission may be copyrighted. According to PLOSu2019s copyright policy, authors who use figures or other material (e.g., graphics, clipart, maps) from another author or copyright holder must demonstrate or obtain permission to publish this material under the Creative Commons Attribution 4.0 International (CC BY 4.0) License used by PLOS journals. Please closely review the details of PLOSu2019s copyright requirements here: PLOS Licenses and Copyright. If you need to request permissions from a copyright holder, you may use PLOS's Copyright Content Permission form.

Potential Copyright Issues:

i) Figures Figure 1A, 3A, and 4A; Please confirm whether you drew the images / clip-art within the figure panels by hand. If you did not draw the images, please provide a link to the source of the images or icons and their license / terms of use; or written permission from the copyright holder to publish the images or icons under our CC BY 4.0 license. Alternatively, you may replace the images with open source alternatives. See these open source resources you may use to replace images / clip-art:

3) Please amend your detailed Financial Disclosure statement. This is published with the article. It must therefore be completed in full sentences and contain the exact wording you wish to be published. State what role the funders took in the study. If the funders had no role in your study, please state: "The funders had no role in study design, data collection and analysis, decision to publish, or preparation of the manuscript.".

**Reviewers' Comments:**

Reviewer's Responses to Questions

**Key Review Criteria Required for Acceptance?**

**Methods**

-Are the objectives of the study clearly articulated with a clear testable hypothesis stated?

-Is the study design appropriate to address the stated objectives?

-Is the population clearly described and appropriate for the hypothesis being tested?

-Is the sample size sufficient to ensure adequate power to address the hypothesis being tested?

-Were correct statistical analysis used to support conclusions?

-Are there concerns about ethical or regulatory requirements being met?

Reviewer #1: -Are the objectives of the study clearly articulated with a clear testable hypothesis stated? YES

-Is the study design appropriate to address the stated objectives? YES

-Is the population clearly described and appropriate for the hypothesis being tested? in part (see below)

-Is the sample size sufficient to ensure adequate power to address the hypothesis being tested? YES

-Were correct statistical analysis used to support conclusions? YES

-Are there concerns about ethical or regulatory requirements being met? NO

Reviewer #2: (No Response)

Reviewer #3: The methodology is well-structured, but some sections could benefit from additional details for reproducibility.

The ethics approval and consent process are mentioned (Page 10, Line 516–519). However, consider including specific details about patient inclusion/exclusion criteria.

On Page 11 (Lines 520–531), the description of sample storage is clear. Adding information about how serum integrity was verified post-transportation would improve this section.

On Page 14 (Lines 550–556), the statistical methods used are appropriate. However, clarify whether any corrections for multiple comparisons were applied during the analysis.

For the whole-genome sequencing (Page 13, Lines 540–549), provide specifics about sequencing depth and coverage uniformity, as these are critical for assessing data quality.

While the manuscript mentions 4045 samples (Page 11, Line 526), justify the sample size for statistical robustness and clinical relevance.

Expand on data normalization or preprocessing steps for variant analysis.

**Results**

-Does the analysis presented match the analysis plan?

-Are the results clearly and completely presented?

-Are the figures (Tables, Images) of sufficient quality for clarity?

Reviewer #1: -Does the analysis presented match the analysis plan? YES

-Are the results clearly and completely presented? YES

-Are the figures (Tables, Images) of sufficient quality for clarity? YES

Reviewer #2: (No Response)

Reviewer #3: Figures and tables are detailed, but some captions lack clarity. For instance, Figure 1 (Page 17) needs an expanded description to guide interpretation.

Results on Page 19 (Lines 600–610) describing SNV distribution could be better supported by including a visual density plot.

The identification of hotspot regions in NS5 and E genes (Page 22, Line 645) is significant but requires contextualization. How do these compare with findings from other regions or similar studies?

Page 23 (Lines 670–675) mentions clustering of severe-associated SNVs in specific genomic regions. Highlight potential functional implications or experimental validations, if any.

The data linking SNVs to disease severity (Page 24, Lines 680–690) are compelling but would benefit from cross-referencing to similar global studies.

Add a supplementary table for the SNVs with their genomic positions, allele frequencies, and clinical correlations.

Consider reordering some subsections for logical flow (e.g., SNV identification before phenotype association).

**Conclusions**

-Are the conclusions supported by the data presented?

-Are the limitations of analysis clearly described?

-Do the authors discuss how these data can be helpful to advance our understanding of the topic under study?

-Is public health relevance addressed?

Reviewer #1: -Are the conclusions supported by the data presented? Partly.

> it needs to be made clear that there was not a single severe dengue case in this study, and the use of the term severe dengue should be clarified, that the study uses this as a self-defined category and refers to pacients with both, thrombocytopenia+leukopenia

-Are the limitations of analysis clearly described?

Patient sample sizes should be mentioned in the results and abstract

-Do the authors discuss how these data can be helpful to advance our understanding of the topic under study?

>YES

-Is public health relevance addressed?

>YES

Reviewer #2: (No Response)

Reviewer #3: The conclusion reiterates key findings effectively. However, some statements (Page 28, Lines 720–725) lack nuance, especially concerning the limitations of the study.

While potential applications of SNVs in clinical practice are mentioned (Page 29, Lines 730–735), more specific examples would add value.

Compare findings with the current state of knowledge to emphasize novel contributions.

Provide actionable steps for future research or surveillance programs.

**Editorial and Data Presentation Modifications?**

Reviewer #1: Table 1: Please provide sample size of patients in table.

Reviewer #2: (No Response)

Reviewer #3: (No Response)

**Summary and General Comments**

Reviewer #1: The article by Ravi et al. adressess an important research question, as to whether genetic variation in the DENV-2 genome is associated with an increased risk of developing severe complications during an infection with the virus. The sample size is large and the clinical characterization of the patient goups should in theory be sufficient to detect genetic variation that may influenze disease severity.

However, there are a few issues and limitations that may be adressed in order to better support the findings and highlight the strengths of the study:

1. It is known that mutations in the 3'UTR of the viral genome, a region that codes for sfRNAs is relevant for viral pathogenesis and immune evasion.

The following two publications provide the evidence for that and should be included in the discussion section of the article:

https://www.ncbi.nlm.nih.gov/pmc/articles/PMC6545344/

https://pubmed.ncbi.nlm.nih.gov/26138103/

I would suggest that the authors specifically looked in more detail for mutations in the relevant regions in the 3'UTR, as they found a significant difference in frequencies of SNVs in the 3'UTR when comparing mild and severe groups (See results section: lines 346 and 351).

2. It is not clear how the authors categorized SNVs in the untranslated regions of the Dengue genome into synonymous and non-synonymous SNVs (mainly Supplementary Table 6). Please clarify.

3. The definitions for mild, severe and moderate Dengue are critical for the entire analysis in this study. This is only mentioned in the methods section. There is a high risk of confusing this with the clinical term of "severe dengue" as based on clincial severe symptoms (see https://www.who.int/emergencies/outbreak-toolkit/disease-outbreak-toolboxes/dengue-outbreak-toolbox).

I suggest to use the definition as explained in lines 545ff throughout the manuscript in order to make clear which groups were analyzed here. Please include sample sizes as well.

4. In terms of disease progression it is important to know at which time point after onset of disease the blood parameter were assessed. As it may be possible that a patient at time of enrolement was categorized as mild or moderate, may later develop thrombocytopenia and leukopenia and fall the "severe group". How can be made sure that this is not the case? Do the authors have any additional information on the outcome of the infection in the enrolled patients?

5. Lines 462ff: The authors state "Critically, the significant alterations observed in hematological and hepatic parameters across the dengue patients highlight their potential as diagnostic and prognostic biomarkers for severe dengue".

The study itself has the potential to do so, and a result may be a list of these biomarkers. I recommend listing them, however, not using the term "severe dengue" as it might be missleading, but instead try to provide biomarkers for thrombocytopenia and leukopenia. Ideally create new subgroups and compare level of hematological and hepatic parameters between the following groups of patients: none (Reference group), thrombocytopenia only, leukopenia only, thrombocytopenia + leukopenia.

Reviewer #2: Ravi et al. collected 4045 serum samples of DENV-NS1-antigen positive and found 3702 DENV-2. Based on 3254 DENV-2 E region sequences and some of full -length sequences, the authors detected nucleotide polymorphisms.

The main finding should be that the sequence variation which is related to disease severity from the title of this manuscript, but it is not clear due to the lack of functional analysis of mutated virus.

Major concerns

1. The authors display the number of SNV, but the specific position or amino acid variation is more important. In this manuscript, the data presentation should be more focused on the chapter “Distinct Genomic Hotspots in E, NS4B and NS5 Regions Underpin Dengue Disease Severity”. In this version, it is not easy to specify the position which is related to disease severity.

2. In line 318, the authors said that “The frequency of the SNVs across the mild, moderate, and severe is depicted in Figure 4D“. It should be shown in Table. The supplemental Tables should show the frequency but not number in three groups, mild, moderate and severe and ANOVA test are required. The escalation of the frequency from mild, moderate, to severe cases will be shown on the responsible SNVs.

3. The Figure 5 must be more conclusive, which display only the position which is related to disease severity. The data presentation of Figure 5 should be re-considered to merge mild, and severe in one line to compare these groups.

4. It is not easy to understand the definition of mild, moderate and severe. The table of criteria should be included. It should be noted that the disease severity of dengue cases are usually determined by clinical symptoms but not the laboratory data.

Minor points:

1. In Figure 2, the color of Mild(M), moderate (MD), Severe (SV) are not distinguishable. It should be modified. For example, decrease the number of reference sequence from the world and enlarge the part of in-house.

2. In line 558, how many sequences were successfully determined whole genome sequence?

Reviewer #3: The manuscript presents a robust clinico-genomic study on DENV-2 with an impressive sample size and advanced sequencing techniques.

Some sections, particularly Results and Conclusions, require tighter integration of findings with broader implications.

PLOS authors have the option to publish the peer review history of their article (what does this mean? ). If published, this will include your full peer review and any attached files.

**Do you want your identity to be public for this peer review?** For information about this choice, including consent withdrawal, please see our Privacy Policy .

Reviewer #1: No

Reviewer #2: No

Reviewer #3: No

**Figure resubmission:**
---

## [Decision Letter · Decision Letter 1]

25 Mar 2025

PNTD-D-24-01925R1Genomic hotspots in the DENV-2 serotype (E, NS4B, and NS5 genes) are associated with Dengue Disease Severity in the Endemic Region of IndiaPLOS Neglected Tropical DiseasesDear Dr. Pandey, Thank you for submitting your manuscript to PLOS Neglected Tropical Diseases. After careful consideration, we feel that it has merit but does not fully meet PLOS Neglected Tropical Diseases's publication criteria as it currently stands. Therefore, we invite you to submit a revised version of the manuscript that addresses the points raised during the review process. Please submit your revised manuscript within 30 days Apr 24 2025 11:59PM. If you will need more time than this to complete your revisions, please reply to this message or contact the journal office at plosntds@plos.org. Please include the following items when submitting your revised manuscript: * A rebuttal letter that responds to each point raised by the editor and reviewer(s). You should upload this letter as a separate file labeled 'Response to Reviewers '. This file does not need to include responses to any formatting updates and technical items listed in the 'Journal Requirements' section below. * A marked-up copy of your manuscript that highlights changes made to the original version. You should upload this as a separate file labeled 'Revised Manuscript with Track Changes '. * An unmarked version of your revised paper without tracked changes. You should upload this as a separate file labeled 'Manuscript '. If you would like to make changes to your financial disclosure, competing interests statement, or data availability statement, please make these updates within the submission form at the time of resubmission. Guidelines for resubmitting your figure files are available below the reviewer comments at the end of this letter. We look forward to receiving your revised manuscript. Kind regards, Zach N AdelmanGuest EditorPLOS Neglected Tropical Diseases David SafronetzSection EditorPLOS Neglected Tropical Diseases

Shaden Kamhawi

co-Editor-in-Chief

Paul Brindley

co-Editor-in-Chief

 **Journal Requirements:**

At this stage, the following Authors/Authors require contributions: Varsha Ravi, Kriti Khare, Ramakant Mohite, Pallavi Mishra, Sayanti Halder, Richa Shukla, Chinky Shiu Chen Liu, Aanchal Yadav, Jyoti Soni, Kanika Kanika, Komal Chaudhary, Neha Neha, Bansidhar Tarai, Sandeep Budhiraja, Pooja Khosla, Tavpritesh Sethi, Md Imran, and Rajesh Kumar Pandey. Please ensure that the full contributions of each author are acknowledged in the "Add/Edit/Remove Authors" section of our submission form.

2) Please ensure that the funders and grant numbers match between the Financial Disclosure field and the Funding Information tab in your submission form. Note that the funders must be provided in the same order in both places as well. State the initials, alongside each funding source, of each author to receive each grant. For example: "This work was supported by the National Institutes of Health (####### to AM; ###### to CJ) and the National Science Foundation (###### to AM).".

**Reviewers' comments:** Reviewer's Responses to Questions

**Key Review Criteria Required for Acceptance?**

**Methods:**

-Are the objectives of the study clearly articulated with a clear testable hypothesis stated?

-Is the study design appropriate to address the stated objectives?

-Is the population clearly described and appropriate for the hypothesis being tested?

-Is the sample size sufficient to ensure adequate power to address the hypothesis being tested?

-Were correct statistical analysis used to support conclusions?

-Are there concerns about ethical or regulatory requirements being met?

Reviewer #1: (No Response)

Reviewer #2: (No Response)

Reviewer #3: There are no further comments

**Results:**

-Does the analysis presented match the analysis plan?

-Are the results clearly and completely presented?

-Are the figures (Tables, Images) of sufficient quality for clarity?

Reviewer #1: (No Response)

Reviewer #2: In response to the reviewer's comment, the authors added one paragraph on the effect of mutation on protein function at lines 504-521. However, among the seven mutations in DI-DIII regions, only C2073T changes amino acid. Similarly, among the six mutations in D3 region, only A2105G changes amino acid. Discussion of the effect of synonymous substitutions on protein function is rather ridiculous and need to be corrected.

Reviewer #3: There are no further comments

**Conclusions:**

-Are the conclusions supported by the data presented?

-Are the limitations of analysis clearly described?

-Do the authors discuss how these data can be helpful to advance our understanding of the topic under study?

-Is public health relevance addressed?

Reviewer #1: 1.Please revise the added part in the new version of the manuscript (lines 669-673).

“While SNVs in the 3′UTR and their epidemiological roles are well studied, our findings highlight

distinct evolutionary pressures in dengue virus adaptation. The mild fNR2 (flaviviral nucleaseresistant

RNA) mutation (A10424G) helps maintain sfRNA (subgenomic flavivirus RNA) stability

for efficient transmission (https://doi.org/10.1016/j.isci.2019.05.019), while the clinically severe

sfRNA mutation (G10296A) may enhance immune evasion by altering interactions with the host

factors like TRIM25 (https://doi.org/10.1126/science.aab3369). This suggests a potential balance

between viral fitness and immune adaptation, influencing disease severity.”

Since there was no functional characterization of the SNVs described to be associated, you may not state "The mild fNR2 mutation helps maintain sfRNA stability...

Please rephrase, this could potentially be true, but at the moment, no evidence is shown.

Please avoid expressions like ..the mild mutation or "the clinically severe mutation".. It would be scientifically better to write " a variant A or B is associated with mild or severe characteristics of a disease, but mutation themselves cannot be mild or severe.

2. Your new finding that the " liver enzyme, AST level observed significantly high

in the dengue patients group having thrombocytopenia and both thrombocytopenia and leukopenia,

suggesting AST level in dengue patient could potentially be used as the prognostic marker for both

thrombocytopenia and leukopenia." should be included in the manuscript.

Reviewer #2: (No Response)

Reviewer #3: There are no further comments

**Editorial and Data Presentation Modifications?**

Reviewer #1: (No Response)

Reviewer #2: (No Response)

Reviewer #3: (No Response)

**Summary and General Comments:**

Reviewer #1: (No Response)

Reviewer #2: (No Response)

Reviewer #3: (No Response)

PLOS authors have the option to publish the peer review history of their article (what does this mean? ). If published, this will include your full peer review and any attached files.

**Do you want your identity to be public for this peer review?** For information about this choice, including consent withdrawal, please see our Privacy Policy .

Reviewer #1: No

Reviewer #2: No

Reviewer #3: No

---

## [Editor Report · Decision Letter 2]

4 Apr 2025

Dear Dr. Pandey,

We are pleased to inform you that your manuscript 'Genomic hotspots in the DENV-2 serotype (E, NS4B, and NS5 genes) are associated with Dengue Disease Severity in the Endemic Region of India' has been provisionally accepted for publication in PLOS Neglected Tropical Diseases.

Best regards,

Zach N Adelman

Guest Editor

David Safronetz

Section Editor

Shaden Kamhawi

co-Editor-in-Chief

Paul Brindley

co-Editor-in-Chief

---

## [Editor Report · Acceptance letter]

Dear Dr. Pandey,

We are delighted to inform you that your manuscript, "Genomic hotspots in the DENV-2 serotype (E, NS4B, and NS5 genes) are associated with Dengue Disease Severity in the Endemic Region of India," has been formally accepted for publication in PLOS Neglected Tropical Diseases.

Best regards,

Shaden Kamhawi

co-Editor-in-Chief

Paul Brindley

co-Editor-in-Chief
